# Single-Cell Analyses of a Novel Mouse Urothelial Carcinoma Model Reveal a Role of Tumor-Associated Macrophages in Response to Anti-PD-1 Therapy

**DOI:** 10.3390/cancers14102511

**Published:** 2022-05-19

**Authors:** Dongbo Xu, Li Wang, Kyle Wieczorek, Yali Zhang, Zinian Wang, Jianmin Wang, Bo Xu, Prashant K. Singh, Yanqing Wang, Xiaojing Zhang, Yue Wu, Gary J. Smith, Kristopher Attwood, Yuesheng Zhang, David W. Goodrich, Qiang Li

**Affiliations:** 1Department of Urology, Roswell Park Comprehensive Cancer Center, Buffalo, NY 14203, USA; dongbo.xu@roswellpark.org (D.X.); li.wang@roswellpark.org (L.W.); kyle.wieczorek@roswellpark.org (K.W.); yue.wu@roswellpark.org (Y.W.); gary.smith@roswellpark.org (G.J.S.); 2Department of Biostatistics & Bioinformatics, Roswell Park Comprehensive Cancer Center, Buffalo, NY 14203, USA; yali.zhang@roswellpark.org (Y.Z.); jianmin.wang@roswellpark.org (J.W.); kristopher.attwood@roswellpark.org (K.A.); 3Departments of Cancer Prevention and Control, Roswell Park Comprehensive Cancer Center, Buffalo, NY 14203, USA; zinian.wang@roswellpark.org; 4Departments of Pathology, Roswell Park Comprehensive Cancer Center, Buffalo, NY 14203, USA; bo.xu@roswellpark.org; 5Departments of Cancer Genetics & Genomics, Roswell Park Comprehensive Cancer Center, Buffalo, NY 14203, USA; prashant.singh@roswellpark.org; 6Department of Pharmacology & Therapeutics, Roswell Park Comprehensive Cancer Center, Buffalo, NY 14203, USA; yanqing.wang@roswellpark.org (Y.W.); xiaojing.zhang@roswellpark.org (X.Z.); yuesheng.zhang@roswellpark.org (Y.Z.); david.goodrich@roswellpark.org (D.W.G.)

**Keywords:** single-cell analyses, mouse urothelial carcinoma, mouse model, ICI immunotherapy, anti-PD-1, tumor-associated macrophages

## Abstract

**Simple Summary:**

Immune checkpoint inhibitors (ICI) have been standard care for advanced bladder cancer patients who fail or are not eligible for chemotherapy. There is an urgent need for preclinical models to study immunotherapy responses because the majority of patients with advanced bladder cancer do not respond to ICI therapy. To develop an authentic bladder cancer preclinical model in immunocompetent mice, we generated a stable organoid and xenograft model by the ex vivo transduction of adenovirus-expressing Cre recombinase into normal mouse urothelial organoid cells isolated from mice engineered with *LoxP* sites flanking the *Trp53*, *Pten*, and *Rb1* genes. The triple knockout (TKO) urothelial organoids developed into high-grade urothelial carcinomas of the basal subtype, both in vitro and in vivo. To study ICI anti-PD-1 treatment responses, the TKO tumors were treated with an anti-PD-1 antibody or a control IgG2a. A mixed pattern of treatment responses was observed. Single-cell analyses of immune cells revealed significantly different infiltration of immune cells between non-responders and responders. A higher percentage of immune cell infiltration, including macrophage and T cell tumor infiltration, was detected in responders compared to non-responders. Overall, these findings suggest that this preclinical TKO model will be a useful tool to study the factors influencing bladder cancer immunotherapy responses and suggest that modulating tumor-associated macrophages may help overcome ICI immunotherapy resistance.

**Abstract:**

Approximately 80% of patients with advanced bladder cancer do not respond to immune checkpoint inhibitor (ICI) immunotherapy. Therefore, there is an urgent unmet need to develop clinically relevant preclinical models so that factors governing immunotherapy responses can be studied in immunocompetent mice. We developed a line of mouse triple knockout (TKO: *Trp53*, *Pten*, *Rb1*) urothelial carcinoma organoids transplanted into immunocompetent mice. These bladder tumors recapitulate the molecular phenotypes and heterogeneous immunotherapy responses observed in human bladder cancers. The TKO organoids were characterized in vivo and in vitro and compared to the widely used MB49 murine bladder cancer model. RNAseq analysis of the TKO tumors demonstrated a basal subtype. The TKO xenografts demonstrated the expression of urothelial markers (CK5, CK7, GATA3, and p63), whereas MB49 subcutaneous xenografts did not express urothelial markers. Anti-PD-1 immunotherapy resulted in a mixed pattern of treatment responses for individual tumors. Eight immune cell types were identified (basophils, B cells, dendritic cells, macrophages, monocytes, neutrophils, NK cells, and T cells) in ICI-treated xenografts. Responder xenografts displayed significantly increased immune cell infiltration (15.3%, 742 immune cells/4861 total cells) compared to the non-responder tumors (10.1%, 452 immune cells/4459 total cells, Fisher Exact Test *p* < 0.0001). Specifically, there were more T cells (1.0% vs. 0.4%, *p* = 0.002) and macrophages (8.6% vs. 6.4%, *p* = 0.0002) in responder xenografts than in non-responder xenografts. In conclusion, we have developed a novel preclinical model that exhibits a mixed pattern of response to anti-PD-1 immunotherapy. The higher percentage of macrophage tumor infiltration in responders suggests a potential role for the innate immune microenvironment in regulating ICI treatment responses.

## 1. Introduction

Bladder cancer is the fourth most common cancer in men in the US and affects more than 80,000 people in the United States annually [1]. More than 90% of bladder cancers are urothelial carcinomas (UC) that originate from the epithelial layer of the bladder [2]. Bladder cancer is classified into two pathological categories: non-muscle invasive bladder cancer (NMIBC) or muscle-invasive bladder cancer (MIBC), according to the depth of tumor invasion. Bladder cancer has been shown to be immunogenic and responds to immunotherapy. Intravesical treatment with immunogenic Bacillus Calmette–Guerin (BCG), for example, has been standard care to treat NMIBC for decades. Recently, FDA-approved immune checkpoint inhibitor (ICI) therapy was approved for targeting programmed cell death protein 1 (PD-1) or programmed cell death ligand 1 (PD-L1) for BCG unresponsive and advanced MIBCs [3,4]. However, the overall response rate for ICI therapy is only approximately 20% [5]. In order to improve the response rate, there is an urgent unmet need to develop relevant preclinical models for studying the mechanisms that influence immunotherapy resistance and response rates.

Genetically engineered and carcinogen-induced mouse models of bladder cancer have potential for investigating mechanisms governing cancer immune response and identifying approaches to overcome treatment resistance. However, few of these models faithfully recapitulate human bladder cancer biology, progression, and treatment responses [6,7,8]. Further, the use of autochthonous mouse models is time-consuming and costly. Transplant models in immunocompetent mice can overcome this limitation. Current immunocompetent mouse transplant models of bladder cancer consist of the MB49 cell line (carcinogen-induced) [9] and the MBT2 cell line (carcinogen-induced) [10]. The most commonly used mouse transplant bladder cancer model involves the injection of the MB49 cell line into the bladder of syngeneic immunocompetent mice intravesically (into bladder lumen) or orthotopically (into bladder wall) [11]. MB49 is a murine bladder cancer cell line derived from C57 BL/6 mice after the exposure of primary bladder epithelial cells to 7,12-dimethylbenz[a]anthracene (DMBA) [9]. However, the molecular characterization of MB49 cells and tumors demonstrates that they have a mesenchymal phenotype closely resembling fibroblasts and do not recapitulate urothelial tumor phenotypes because they do not express EpCAM, Keratin 5, or Keratin 14 [12]. Further, exome sequencing of N-butyl-N-(4-hydroxybutyl)-nitrosamine (BBN)-induced bladder tumors revealed significant tumor heterogeneity due to variable mutation profiles and neoantigens in individual tumors [7,12]. Alternatively, normal human and mouse epithelial cells can be genetically engineered ex vivo to quickly establish tumor models [13,14,15]. We used such an approach to develop a mouse model that mimics the molecular forces that drive aggressive human MIBC, by inactivating three tumor suppressors (*Trp53, Pten,* and *Rb1*) in normal urothelial cells. The P53, PI3K/AKT, and RB1 pathways are among the most commonly altered pathways in human MIBC [16]. We successfully generated a mouse TKO (triple knockout) organoid that develops into urothelial carcinomas upon transplantation into immunocompetent C57 BL/6J mice. The molecular subtype and the immunotherapy responses of the TKO xenografts were characterized. Results indicate these TKO tumors provide a preclinical experimental model that recapitulates clinically relevant bladder cancer histology and immunotherapy responses useful for further delineating the factors influencing response rates to ICI immunotherapy.

## 2. Materials and Methods

### 2.1. Isolation of Primary Urothelial Cells

Triple-floxed mice (*Trp53 ^f/f^*, *Pten ^f/f^*, *Rb1^f/f^*) were reported previously [17]. The construction and genotyping of this genetically engineered mouse have been described previously [18,19,20,21]. To create a strain compatible with C57BL/6 mice, mixed background triple-floxed mice (C57BL/6:129/Sv:FVB) were bred with homogenized C57BL/6 background mice for multiple generations. The process of urothelial cells disassociation and ex vivo transduction was described [22]. Briefly, four fresh bladder tissues were collected from male triple-floxed mice for disassociation. Each was sliced into four pieces, and all pieces were mixed and incubated in 5 mL of the culture medium without charcoal-stripped FBS and supplemented with 1:10 dilution of collagenase/hyaluronidase (STEMCELL Technologies) at 37 °C for 30 min. The complete culture medium was comprised of Mammary Epithelial Cell Growth medium supplemented with the provided growth factors (Lonza CC-3150) and 5% charcoal-stripped FBS (Thermo Fisher), 1% GlutaMAX (Gibco), 100 µg/mL Primocin (InvivoGen ant-pm-1), and 10 µM fresh Y-27632 (Selleckchem). Then, disassociated cells were collected and resuspended in 2 mL TrypLE™ Express Enzyme (Gibco) for 3 min at room temperature. Then, 5 mL of the complete culture medium was added, and the dissociated cells were passed through a 100 μm sterile cell strainer.

### 2.2. Adenoviral Transduction and Organoid Culturing

An adenovirus with a CMV promoter driving the expression of Cre recombinase was obtained from the University of Iowa Vector Core Facility (Ad5CMVCre, VVC-U of Iowa-5-HT). The dissociated cells were resuspended in 0.5 mL complete culture medium in 24 well plate and mixed with 2 × 10^7^ PFU of adenovirus. The plate was centrifuged for 30 min at 300× *g*. Next, the plate was put into an incubator at 37 °C for 1 h. Cells were collected and resuspended in 70% Matrigel (Corning)/complete culture medium and plated in pre-warmed 6-well culture plates for TKO organoids. Organoid passaging and freezing were performed as reported [23].

### 2.3. PCR Validation

Genome DNA was extracted from the TKO organoid cells. PCRs (KAPA Taq PCR kit) were performed to detect wild-type (WT), floxed, or Cre recombined alleles of *Trp53*, *Rb1,* and *Pten*. PCR results were analyzed by agarose gel electrophoresis stained with ethidium bromide. PCR primers were listed in Appendix A.

### 2.4. Histology, Immunohistochemistry, and Immunofluorescence

Mice were euthanized for tumor tissue collection when moribund. The bladder along with the urethra was fixed in phosphate-buffered 4% paraformaldehyde and embedded in paraffin. Hematoxylin and eosin (H&E) and immunostaining were performed on 5 μm thickness sections. For the organoid sections, the TKO cell pellets were embedded into paraffin using HistoGel (Thermo fisher) [24]. Immunofluorescent (IF) staining and immunohistochemistry (IHC) were performed as previously described [25]. Primary and secondary antibodies used were listed in Appendix A. Histology and IHC immunostaining were reviewed and verified by a pathologist (B.X.).

### 2.5. Transcriptional Profiling of Urothelial Tumors

Total RNA was extracted from frozen bladder tumors (20 mg) using Qiazol reagent and miRNeasy Mini Kit (Qiagen). The sequencing libraries were prepared with the RNA HyperPrep Kit with RiboErase (HMR) (Roche Sequencing Solutions), from 500 ng total RNA. Libraries were pooled and sequenced on an Illumina NovaSeq 6000 following the manufacturer’s protocol. Sequencing reads that passed the quality filter from Illumina RTA were first processed using FASTQC (v0.10.1) for sequencing base quality control. Then, sample reads were aligned to the human reference genome (GRCm38) and a GENCODE (version 22) annotation database using STAR2 [26]. A second round of QC using RSeQC [27] was applied to mapped bam files to identify potential RNASeq library preparation problems. Gene level raw counts were obtained using Subread package [28]. Differential gene expression analysis was performed using DESeq2 [29], and pathway analysis was performed with the Gene Set Enrichment Analysis (GSEA) method (4.0) [30]. The GSEA tool was chosen to run the analysis using the normalized gene count data that pre-filtered the low count genes. Pathway analysis was run against MSigDB, a collection of annotated and curated gene set repositories offered by the developer of GSEA (Broad Institute MIT and Harvard). This particular run used a database, which is C2 of version 7.4 collection, containing 2307 gene sets from various well-known and up-to-date pathway databases such as BioCarta, KEGG, and Reactome, among others. QIAGEN ingenuity pathway analysis (IPA) [31] was performed in observation between TKO intravesical tumors and MB49 intravesical tumors by using the threshold of *p* < 0.05 and *padj* < 0.05, and fold change > 2, while it was performed in observation between TKO intravesical tumors and normal female WT cells by using the threshold of *p* < 0.05 and *padj* < 0.05, and fold change > 1.5.

### 2.6. Intravesical and Orthotopic Injections and Drug Response

All animal procedures were approved by the Institutional Animal Care and Use Committee (IACUC) at Roswell Park Comprehensive Cancer Center (1395M). For orthotopic tumor formation, 2 × 10^6^ TKO tumor cells in 50% Matrigel/medium were injected orthotopically into the bladder wall of C57 BL/6J mice (Jackson Lab, Strain #000664). The bladder was exposed by a midline abdominal incision, and cell suspension was intramurally injected into the wall of the bladder dome using a Hamilton syringe with a 30 G needle (Hamilton 7803-07) [32]. For intravesical tumor formation, 2 × 10^6^ TKO tumor cells in medium were injected into the bladder via a 24 G 3/4″ catheter (Exelint) as described [33]. Briefly, the bladder was catheterized and pretreated with 100 μL 0.1% Poly-L-Lysine for 20 min. 2 × 10^6^ TKO cells were injected into the bladder for 1.5 h dwelling time. The MB49 cell line for intravesical injection was obtained from Dr. Ratha Mahendran, Department of Surgery, National University of Singapore [34]. 

For the drug responses to anti-PD-1 antibody, the TKO organoid cells were subcutaneously injected into immunocompetent C57 BL/6J mice for tumor expansion. The subcutaneous tumors were collected and dissociated into cell suspensions. Then, 2 × 10^6^ cells in 50% Matrigel/medium were subcutaneously injected into the right flank of male C57 BL/6J mice (~8 weeks old). Tumor-bearing mice were randomly grouped into 2 groups (*n* = 9) once tumors reached approximately 120 mm^3^. Mice were treated with an anti-PD-1 antibody (BioXcell, RMP1-14, 200 μg, intraperitoneally twice weekly, Catalog: BP0146) or a control IgG2a (BioXcell, 2A3, 200 μg, intraperitoneally twice weekly, Catalog: BP0089) for 2 weeks. Tumors were measured with a caliper in two dimensions, and volumes were calculated by the formula V = (Width × Width × Length)/2. After two weeks of treatment, tumor tissues were dissected and processed on the next day of the last treatment for paraffin histologic analysis and single-cell dissociations. The response to the treatment was a defined 50% decreased volume compared to the average of control IgG2a treated tumors.

### 2.7. Single-Cell Sequencing

Single-cell dissociations were performed from TKO xenografts treated with control IgG2a (*n* = 2) or anti-PD-1 (responders (*n* = 2) and non-responders (*n* = 2)). Single-cell suspensions of two xenografts were pooled together and assessed with Trypan Blue for quality control (more than 90% live cells). Single-cell libraries are generated using the 10× Genomics Chromium Next GEM Single-Cell 3′ Reagent Kit v3.1, sequenced on an Illumina NovaSeq 6000, as previously reported [35]. For the Chromium 10× Genomics libraries, the raw sequencing data were processed using Cellranger [36] software to generate the fastq files and count matrices. Then, the filtered gene-barcode matrices, which contain barcodes with the Unique Molecular Identifier (UMI) counts that passed the cell detection algorithm, were used for further analysis. The downstream analyses were performed primarily using Seurat [37] single-cell data analysis R package. First, cells with unique feature counts over 5000 or less than 200, or cells that have >10% mitochondrial counts, were filtered out from the analysis to remove dead cells and doublets. Dimension reductions, including principal component analysis (PCA), UMAP, and tSNE, were performed using the highly variable genes. Data clustering was identified using shared nearest neighbor (SNN)-based clustering on the first 30 principal components. Immune cell clusters were identified as Ptprc (CD45)-positive and were further annotated using SingleR [38] package with the ImmGen reference database. The ImmGen database contains microarray gene expression data of over 253 samples, which include the 20 main flow sorted mouse immune cell types, from the immunological genome project. The main cell types include T cells, NK cells, and macrophages, that were identified in our dataset. Instead of using a small set of marker genes only, SingleR labels each cell by similarity with the reference based on whole transcriptome; the similarity calculation is based on the Spearman coefficient of variable genes of each cell with the reference.

### 2.8. Statistical Analysis

GraphPad Prism 8 was used to perform all statistical analyses. Two-tailed unpaired *t*-tests with Welch’s correction were used to compare two groups. One-way analysis of variance (ANOVA) with Tukey’s multiple comparisons test was used to compare more than 2 groups. In all tests, significant differences were indicated by the values of *p* < 0.05 (*), *p* < 0.01 (**), and *p* < 0.001 (***).

## 3. Results

### 3.1. Ex Vivo Inactivation of Trp53/Rb1/Pten Results in High-Grade Urothelial Carcinoma Organoids

The use of autochthonous mouse models is time-consuming, laborious, and expensive. To provide a faster and more tractable model, we employed an ex vivo organoid culture system to delete the *Trp53*, *Pten,* and *Rb1* tumor suppressors from normal mouse urothelium [22]. The genetic inactivation of these three tumor suppressors is relevant because P53/RB1/PTEN pathways are commonly altered in human MIBC [39,40,41]. Normal urothelial cells from the bladders of triple-floxed mice (*Trp53^f/f^: Pten^f/f^: Rb1^f/f^*) were enzymatically disassociated for the generation of the TKO organoids. The unselected cells were transduced ex vivo with adenovirus designed to express Cre recombinase driven by a CMV promoter (Ad5CMVCre) and subjected to organoid culture and in vivo and in vitro characterization (Figure 1A). The TKO organoids could be successfully passaged both in vivo (5 times) and in vitro (8 times). H&E staining of paraffin sections from the TKO organoids demonstrated high-grade urothelial carcinoma with squamous differentiation (Figure 1B). The TKO organoids also showed strong expression of basal cell markers (CK5 and p63) and no expression of luminal cell markers (CK20 and CK8) (Figure 1B). PCR confirmed the complete gene deletions of *Trp53, Pten,* and *Rb1* in the TKO organoids (Figure 1C). These results suggest that our TKO organoids are likely derived from CK5-expressing cells that reside in the intermediate or basal layers.

### 3.2. TKO Is a Basal-like Bladder Cancer Model

Two major subtypes of bladder cancer, luminal or basal, have been identified [42]. Basal tumors are enriched in CK5, CK14, and CD44, whereas luminal cell-like tumors are enriched in uroplakins, CK18, GATA3, and CK20. To characterize the tumorgenicity of the TKO organoids, the TKO cells were first injected subcutaneously (SQ). After tumor formation, subcutaneous tumors were dissociated and passaged for generation as tumor-derived organoids. Recombined *Pten*, *Trp53*, and *Rb1* alleles in subcutaneous tumors and tumor-derived organoids were confirmed by PCR (Figure 1C). To characterize the tumorigenicity of TKO cells in the bladder microenvironment, dissociated TKO cells from the subcutaneous tumors were injected intravesically and orthotopically into C57BL/6J mice. The SQ xenografts showed a 100% success rate and rapid growth in two weeks, whereas the orthotopic and intravesical tumor formations were only observed in 63% (5/8) or 29% (2/7), respectively (Appendix A). Three of the five orthotopic tumors were found to have a muscle-invasive disease (Appendix A). The TKO cells were also injected into the tail veins of 3 C57 BL/6J mice and developed 100% (3/3) lung metastasis (Appendix A). To further examine luminal and basal cell types, we performed IHC/IF analyses of urothelial marker expression in the TKO intravesical tumors. In parallel, the MB49 cells were injected intravesically as a reference for comparison. MB49 is the most commonly used syngeneic mouse bladder cancer cell line. The intravesical TKO tumors displayed the morphology of high-grade urothelial carcinoma and squamous differentiation (Figure 2). The TKO tumors revealed positive protein expression of urothelial lineage markers (CK5, CK14, CK7, GATA3, and p63) and negative expression of Uroplakin 3, CK20. The expression pattern of urothelial markers in intravesical tumors is identical to subcutaneous (published in [22]) and orthotopic tumors (Appendix A). In contrast, the MB49 tumors expressed no urothelial markers. 

To assess the similarities between the TKO models and MB49 cells, whole-transcriptome profiling was performed for 3 MB49 intravesical tumors, 2 TKO intravesical tumors, and 2 TKO orthotopic tumors. Principal component analysis (PCA) across samples demonstrated that MB49 cells had transcriptome profiles that differed significantly from the TKO tumors, and intravesical TKO and orthotopic TKO tumors showed minimal variances (Figure 3A). Both intravesical TKO and orthotopic TKO tumors demonstrate transcriptome profiles that resembled human basal-like MIBC (Figure 3B) using a luminal/basal/neuronal classifier [43]. Further, GSEA analysis indicates the TKO tumors exhibited the downregulation of cell cycle genes and the upregulation of extracellular matrix (ECM) genes compared to MB49 tumors (Appendix A). Ingenuity pathway analysis (IPA) showed that the TKO tumors display basal cell carcinoma signaling activation and PI3K/AKT signaling inhibition compared to the MB49 tumors (Appendix A). The inhibition of cell cycle G1/S checkpoint regulation and the activation of CREB signaling were observed in the TKO tumor compared to normal female wild-type urothelial cells (Appendix A). IPA, including the canonical pathway and an upstream regulator analysis, and GSEA analysis data are summarized in Appendix A. Together, these findings suggest that the TKO tumors closely resemble basal-like human bladder cancer, and MB49 exhibited no expression of urothelial markers, consistent with the previous molecular characterization of MB49 [12].

The heterogeneous growth pattern (take rate) of different models (SQ vs. orthotopic vs. intravesical) may be explained by variable injection techniques, the different levels of efficiency of tumor cell inoculation/adhesion to the bladder wall, or the important role played by the tumor microenvironment. Given the 100% success rate of TKO SQ tumor formation, we further investigated the immune microenvironment of SQ tumors. IHC of CD4, CD8, and PD-1 showed more T cell infiltration in the TKO tumors than in the MB49 tumors (Figure 3C,D). Therefore, the TKO model is a “urothelial-like” and “hot” bladder cancer tumor compared to the “fibroblast-like” and “cold” MB49 model.

### 3.3. TKO Tumors Demonstrate Heterogeneous Responses to Anti-PD-1 Treatment

To determine the immunotherapy response of our TKO model, the subcutaneous TKO tumors were used for anti-PD-1 treatment given the 100% take rate and immune cell infiltration (Figure 3C,D). As expected, anti-PD-1 treatment did decrease the tumor volumes and weight significantly. This response occurred without significant toxicity, as indicated by the minimal mouse weight changes observed (Figure 4A,B, Appendix A) compared with the tumors treated with control IgG2a. However, a mixed pattern of treatment responses was observed in nine individual anti-PD-1 treated tumors. The discrimination between responders and non-responders was determined by observing the segregation of the tumor growth curves and comparing the *p* values between responders and non-responders (Welch’s *t* test). Using a cutoff of 50% reduced tumor volume compared to the average of the control IgG2a-treated tumors (992.03 mm^3^), 6 of 9 were categorized as non-responders and 3 of 9 as responders (Figure 4A). The mixed responses to anti-PD-1 treatment mimic the clinic response rate (20–30%) in bladder cancer patients treated with immunotherapy. 

### 3.4. Anti-PD-1 Treatment Responders Exhibit Increased Immune Cell Infiltration 

To determine the mechanism of differential responses to anti-PD-1 therapy, single-cell whole transcriptome profiling for tumor xenografts from control IgG2a (*n* = 2), responders (*n* = 2), and non-responders (*n* = 2) were performed. Cells from xenograft tumors of control IgG2a treatment (*n* = 2, 4762 cells), anti-PD-1 non-responders (*n* = 2, 4459 cells), and anti-PD-1 responders (*n* = 2, 4861) were annotated using SingleR with the ImmGen reference database, and 15 major different cell types were identified (Appendix A). Single-gene feature plots in 2 control IgG2a tumors showed positive expression of CK5, TP63, CK14, GATA3, and CK7 and negative expression of Upk3 and CK20 (Appendix A) in the TKO tumor cells, consistent with the IHC/IF expression in Figure 2. Re-analyzing tumor cells selected for high EpCAM expression revealed 11 different clusters, revealing heterogeneity within tumor cells (Appendix A). Next, immune cell subset analyses were performed using CD45 as an immune cell marker and renormalized. Eight clusters of immune cells were identified (Appendix A). Further annotation using SingleR with ImmGen revealed cell populations of basophils, B cells, dendritic cells, macrophages, monocytes, neutrophils, NK cells, and T cells in anti-PD-1 treated xenografts (Figure 4C and Table 1). A clustering heatmap of immune cell types in the TKO tumors across the top 10 most differentially expressed genes were shown in Appendix A, and Violin plots of representative immune cell markers in macrophage (Cd68), T cells (Cd3g), and NKT cells (Nkg7) were shown in Appendix A. Responder tumors displayed significantly increased immune cell infiltration (15.3%, 742 immune cells/4861 total cells) compared to the non-responder tumors (10.1%, 452 immune cells/4459 total cells, Fisher Exact Test *p* < 0.0001). Specifically, there were more T cells (46/4861 vs. 18/4459, *p* = 0.002) and macrophages (420/4861 vs. 287/4459, *p* = 0.0002) in responder xenografts than in non-responder xenografts (Figure 4D). The TKO tumors treated with anti-PD-1 showed significantly fewer PD-1 positive cells in IHC compared to TKO tumors treated with control IgG2a, suggesting successful PD-1 blockade in both responders and non-responders. Compared to control IgG2a tumors, responding tumors exhibited an immune-inflamed phenotype with significant infiltration of T cells (CD4 and CD8 markers) and macrophages (F4/80 marker), whereas non-responder tumors showed no significant change in T cells or macrophage infiltration (Figure 5A,B). GSEA analysis showed that chemokine receptor and IFN-gamma response-related genes were upregulated in tumor cells in responders compared to tumor cells in non-responders (Appendix A). GSEA of immune cells between responders and non-responders revealed translation-related genes upregulation in macrophages and complement cascade related genes upregulation in T cells in responders (Appendix A). Together, the single-cell data revealed a higher percentage of T cells and macrophages infiltrating tumors in responders, suggesting a potential role for the innate immune microenvironment for anti-PD-1 treatment responses.

## 4. Discussion

PD-1/PD-L1 blockade immunotherapy is approved by the FDA for treatment of advanced bladder urothelial carcinoma. However, only 20% of patients with advanced bladder cancer respond to anti-PD-1/anti–PD-L1 treatment [44]. A major barrier in bladder cancer translational research is the lack of authentic mouse bladder cancer models for studying the factors that influence the response to immunotherapy. An in vivo bladder cancer model with an intact immune system is required for the evaluation of immunotherapeutic responses, and a transplant model would optimize experimental throughput. Here, we present a unique TKO model of basal-like bladder cancer for studying bladder cancer biology and treatment responses in immunocompetent mice. The TKO model combines *Trp53*, *Pten,* and *Rb1* deletion and displays a transcriptional profile similar to human basal-squamous bladder cancer, representing 1/3 to 1/2 of MIBC patients [16]. The TKO model also presents many similarities to the immune microenvironment of human bladder pathology, suggesting that the TKO model is a better preclinical model for studying the immunotherapy response for bladder cancer than other commonly used transplant models.

Most widely used murine transplant models of bladder cancer are derived from bladder tumors developing in carcinogen-treated immunocompetent mice. Such models include MB49 cells that are implanted into syngeneic C57 BL/6J mice or MBT-2 cells implanted into syngeneic C3H/He mice [45,46]. However, MB49 cells were found to closely resemble fibroblasts and did not recapitulate the features of urothelial carcinoma (Figure 3B) [12]. The alternative approach is to initiate autochthonous bladder cancers by delivery of the carcinogen BBN in drinking water. BBN-induced tumors mimic a spectrum of diseases analogous to human MIBC with similar mutation frequencies and signatures [7]. However, the inherent mutational heterogeneity of the BBN model makes it difficult to study the factors influencing anti-tumor immunity because genetic differences will confound results [6,7,12]. Rapid and efficient ex vivo gene editing in organoids has accelerated the pace of generating clinically relevant experimental models [14]. We have used a rapid ex vivo approach by employing the Cre expressing adenovirus to delete clinically relevant tumor suppressor genes (*Trp53*, *Pten*, and *Rb1*) from normal urothelial cells with high efficiency, as described [22]. This ex vivo approach can be applied to any combination of genes of interest to generate preclinical transplant models within a few weeks.

P53, PTEN/PIK3CA, and RB1 are the most common altered pathways in muscle-invasive bladder cancer [16]. Concurrent mutations of *TP53*, *PTEN,* and *RB1* have been reported in bladder cancer patients [47]. The present TKO model generates urothelial tumor in subcutaneous, orthotopic, intravesical, and metastatic (lung) environments (Appendix A), and all these tumors share the characteristics of human basal-squamous bladder cancer (Figure 2 and Figure 3). Of note, the take rate of intravesical and orthotopic models is relatively low (29% and 63%, respectively), possibly due to limited cell adhesion to the bladder wall, short indwelling (1.5 h) time, or the effects of the tumor microenvironment. In any case, the TKO model can be further optimized to increase the take rate (e.g., prolonged contact time, bladder cauterization, and in vivo selection) of the intravesical models [46,48].

The urothelium is composed of a basal cell layer (positive for CK5 and p63), 1–2 layers of intermediate cells (Uroplakins and p63 positive), and umbrella cells (Uroplakins, CK8 and CK20 positive) [6]. Adenovirus Cre with a CMV promoter used in this study may target a wide range of cell types after tissue disassociation (basal vs. luminal, or urothelial vs. non-urothelial). The TKO organoids may thus be derived from basal cells within these cell cultures given the basal-like molecular subtype (Figure 2 and Figure 3B). The ex vivo approach described here with the purification of different cell types (EpCAM/CD49f) prior to adenovirus treatment could be used to definitively identify the cell of origin and study how the cell of origin impacts tumor formation, progression, and immunotherapy responses [49,50,51]. 

The TKO subcutaneous tumors not only model clinically relevant bladder cancer histology but also recapitulate the variable clinical responses to immunotherapy (response rate 20–30%). The variability of immune cell infiltration (Figure 5, T cells, macrophages) may help explain the 33% (3/9) response rate of anti-PD-1 treatment (Figure 4A). The immune cells from host C57 BL/6J mice can be distinguished by PCR (WT alleles versus recombination alleles) from the TKO cells (Figure 1C). Here, we observed increased immune cell infiltration (T cells and macrophages) in the TKO responders compared to the TKO non-responders. The strong infiltration of macrophages in the TKO responders suggests a need for innate immune cells in the tumor microenvironment for an optimal response. Perhaps the host innate immune system exists in a more highly activated state in responder mice compared to the non-responder mice. An alternative explanation is that intrinsically heterogeneous tumor cells (11 clusters) respond differently to anti-PD-1 treatment, although a minimal difference in tumor cells was observed between responders and non-responders (Appendix A). Finally, it is also possible that the mixed genetic background of the TKO cells may impact anti-PD-1 treatment responses in C57 BL/6J mice. However, the 100% (9/9) rate of tumor formation in the control IgG2a group (Figure 4A) and the 100% (3/3) rate of lung metastasis (Appendix A) suggest that the TKO cells are MHC-compatible with C57BL/6 mice.

The immune system in the context of bladder cancer is highly active, with multiple immune cells involved [52]. Tumor-associated macrophages (TAMs) are the most abundant infiltrating immune cells (mean density 14.55 cells/mm^2^) in the tumor core of bladder cancer [53]. Few studies have investigated the effect and mechanisms of TAMs in bladder cancer. Consistent with human bladder cancers, the macrophage is the most abundant of the immune cells (3.95% of all cells) in the TKO tumors, compared to 0.34% NKT cells, 0.31% T cells, and < 0.01% B cells in the mouse TKO tumors (Table 1). Further flow cytometry studies using CD45, CD11b, and F4/80 markers will quantify the absolute number of immune cells and macrophages in responders and non-responders. Anti PD-1 therapies may function through a direct effect on TAMs because both mouse and human TAMs express PD-1 [54] and anti-PD-1 can be transferred to macrophages by Fc-gamma receptors (FcγRs) [55]. More in-depth studies of TAMs are warranted to “redirect macrophage function to sustain their defender antitumor activity” in bladder cancer [56]. New immunotherapeutic strategies involving targeting TAMs are being developed to improve ICI treatment responses [57]. Such strategies could be tested in the context of bladder cancer, using the model we have described here.

## 5. Conclusions

In conclusion, we have generated a transplant mouse model of the basal-squamous subtype of bladder cancer using a quick and efficient ex vivo adenovirus-mediated approach. The TKO tumor resembles a human bladder cancer immune microenvironment and can be used for studying the mechanisms of immune response or resistance. Single-cell analyses of responding tumors reveal a potential role for tumor infiltrating macrophages. Future studies are warranted to determine the causal relationship and mechanisms between TAMs and anti-PD-1 responses.

## Figures and Tables

**Figure 1 cancers-14-02511-f001:**
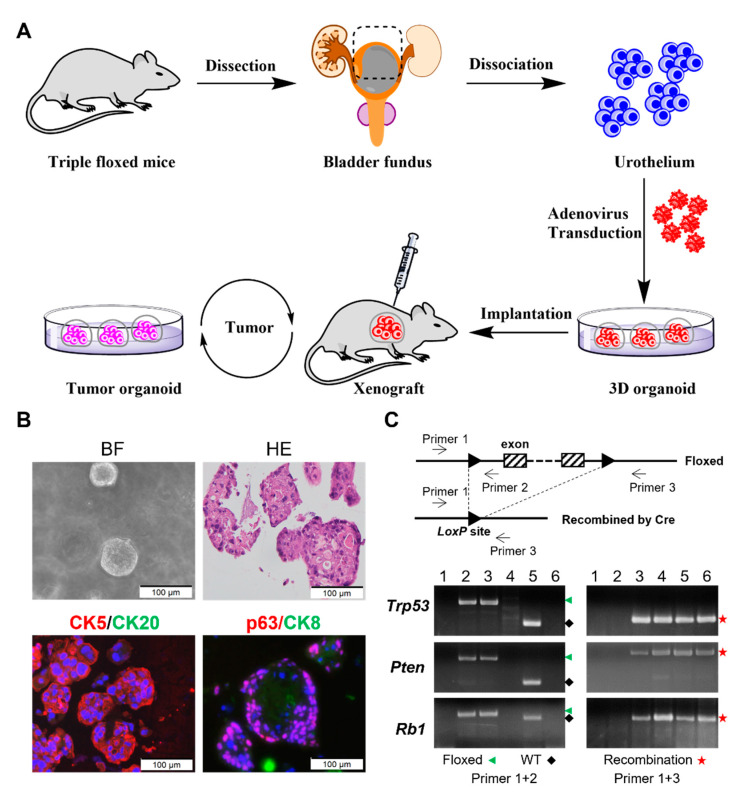
The generation and characterization of TKO organoids created by ex vivo A5CMVCre transduction of urothelial cells. (**A**) The workflow of TKO organoid generation via ex vivo Cre recombination by Ad5CMVCre transduction. (**B**) A bright-field (BF) image of organoids, H&E, and the immunofluorescence staining of organoid paraffin sections. (**C**) The PCR validation of floxed/WT and Cre recombined alleles in indicated samples. PCR bands were stained with ethidium bromide. The genomic structure of floxed allele and recombined allele with *LoxP* sites is shown. Primer 1 and 2 were used for floxed and WT alleles, whereas primer 1 and 3 were used for recombined alleles. Genotyping using triple-floxed mouse tail DNA (lane 2) revealed floxed alleles (green) of *Trp53*, *Pten,* and *Rb1* and no recombination alleles (red). WT alleles (black) were detected only in WT stroma cells from the host in TKO subcutaneous tumor (lane 5) but not detected in TKO primary organoid (lane 4) or tumor derived organoid after 7 serial passages (lane 6). The recombination alleles (red) were detected in the TKO organoid (4), xenograft tumor (5), and tumor-derived organoid (6). A mixture of cells with floxed and recombination alleles served as a positive control (lane 3), and H_2_O served as a negative control (lane 1). Original DNA gel figures with DNA ladders and the expected size of PCR bands are shown in Appendix A.

**Figure 2 cancers-14-02511-f002:**
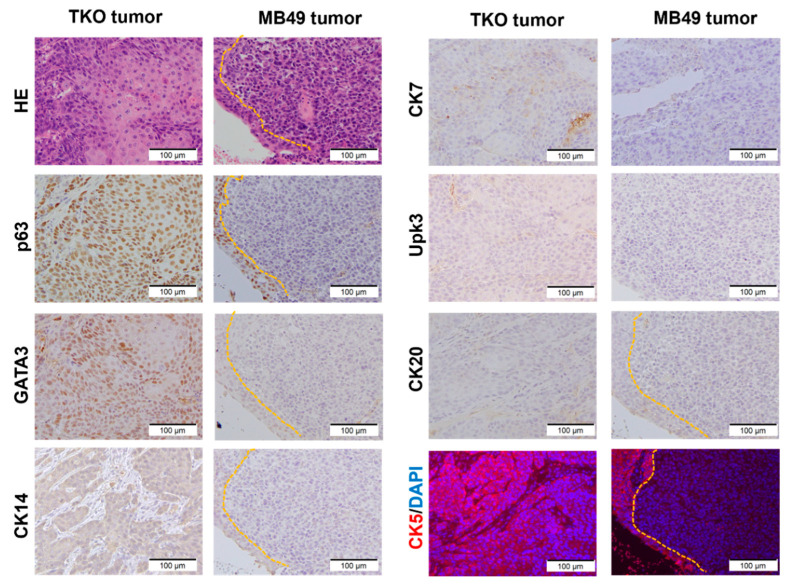
The expression of urothelial markers in TKO intravesical tumors. Sections from the TKO and MB49 intravesical tumors were stained by IHC and IHC-IF with urothelial lineage markers. The TKO tumors are positive for p63, GATA3, CK14 (weak), CK5, and CK7 (patchy), whereas the M49 tumors are negative for urothelial lineage markers. Note the strong positive stain of p63 and CK5 in the normal urothelium adjacent to the MB49 tumors (yellow dash line).

**Figure 3 cancers-14-02511-f003:**
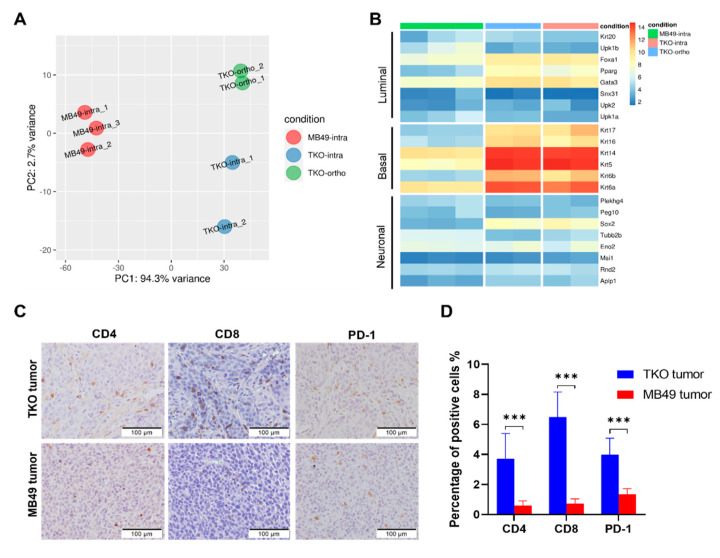
The molecular and immune characterization of TKO tumors. (**A**) The inter-and intra-group variability of the TKO tumors was illustrated by the principal component analysis (PCA) of the RNA-seq data. Each dot represents a tumor: TKO intravesical tumors (TKO-intra, blue), TKO orthotopic tumors (TKO-ortho, green), and MB49 intravesical tumors (MB49-intra, red). (**B**) Clustering heatmap of TKO intravesical tumors (pink), TKO orthotopic tumors (blue), and MB49 intravesical tumors (green) using the luminal/basal/neuronal classifier. (**C**) Representative IHC staining of CD4, CD8, and PD-1 in subcutaneous tumor tissues. (**D**) The quantification of CD4, CD8, and PD-1 positive cells in subcutaneous tumor sections stained by IHC. *p*-value, ANOVA with Tukey’s multiple comparisons test, *p* < 0.001 (***).

**Figure 4 cancers-14-02511-f004:**
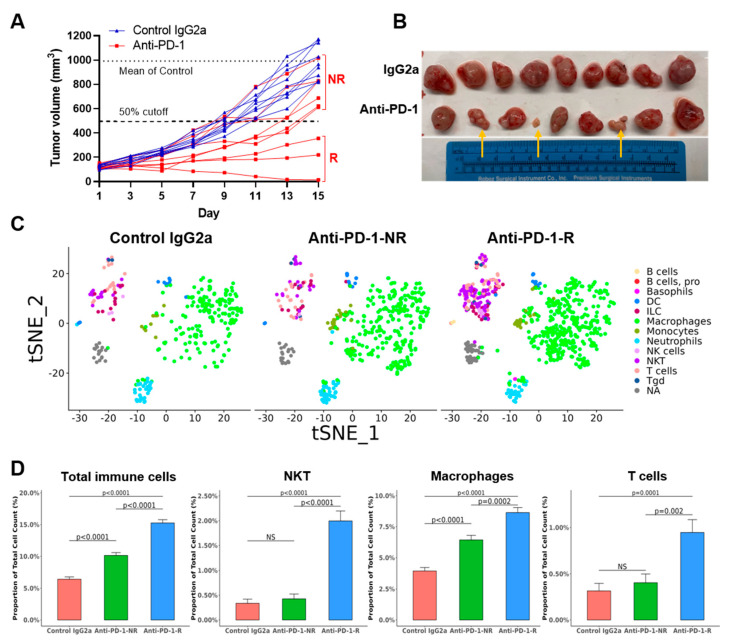
The treatment responses of the TKO tumors to anti-PD-1 in C57 BL/6J mice. (**A**) Individual tumor volume growth curves in C57 mice (*n* = 18) bearing TKO tumors treated with anti-PD-1 (*n* = 9, red) or control IgG2a (*n* = 9, blue). The mean volumes of non-responders (NR) and responders (R) were 765.1 mm^3^ and 196.4 mm^3,^ respectively (*p* = 0.01, Welch’s *t* test). (**B**) The gross tumors of the anti-PD-1 group (lower) and the control group (upper). Anti-PD-1 immunotherapy resulted in three responders (indicated by yellow arrows) vs. six non-responders. (**C**) Immune cells (basophils, B cells, dendritic cells, macrophages, monocytes, neutrophils, NK cells, and T cells) in the TKO tumors are shown in different colors (left: control IgG2a; middle: non-responders; and right: responders). (**D**) The percentage of total immune cells, NKT cells, macrophages, and T cells, in total cells from xenografts, were compared in control tumors, non-responders, and responders. *p*-value, Fisher Exact Test.

**Figure 5 cancers-14-02511-f005:**
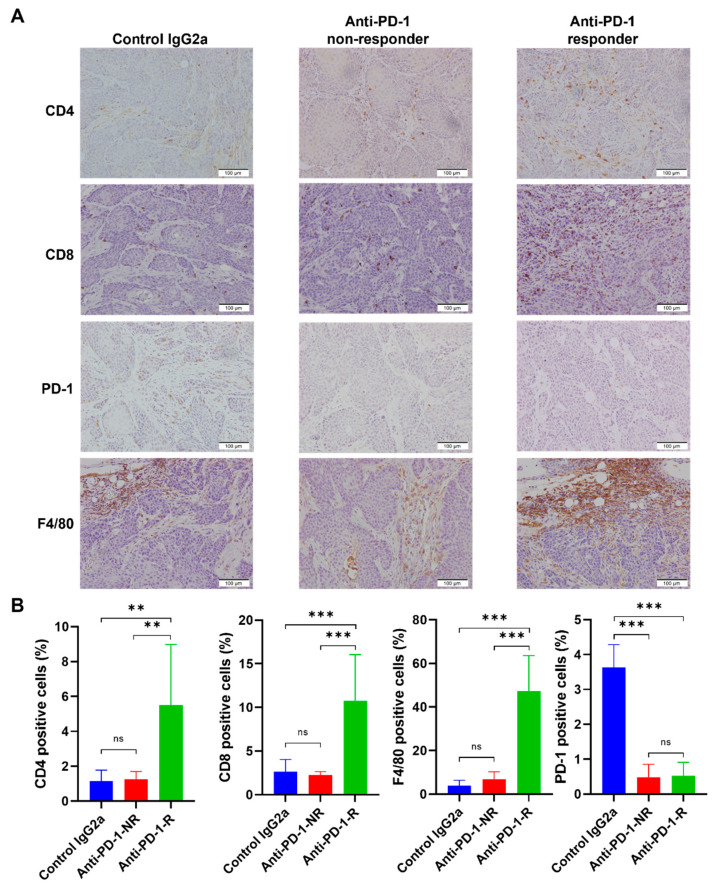
IHC staining of CD4, CD8, PD-1, and F4/80 in control and anti-PD-1 treated TKO tumors. (**A**) Representative IHC staining (CD4, CD8, PD-1, and F4/80) is shown in control IgG2a, anti-PD-1 (non-responders), and anti-PD-1 (responders). (**B**) The quantification of the percentages of CD4, CD8, PD-1, and F4/80 positive cells in the TKO tumors treated with control IgG2a, anti-PD-1 (non-responders), and anti-PD-1 (responders). *p*-value, ANOVA with Tukey’s multiple comparisons test, *p* < 0.01 (**), and *p* < 0.001 (***).

**Table 1 cancers-14-02511-t001:** The cell counts and percentages of the immune subsets in all cells.

Cell Type	Control IgG2a Cell Count	Anti-PD-1-NR Cell Count	Anti-PD-1-R Cell Count	Control IgG2a % in Total Cell	Anti-PD-1-NR % in Total Cell	Anti-PD-1-R % in Total Cell
Basophils	0	0	2	0.00	0.00	0.04
B cells	0	0	3	0.00	0.00	0.06
B cells, pro	0	0	1	0.00	0.00	0.02
DC	14	14	20	0.29	0.31	0.41
ILC	12	15	33	0.25	0.34	0.68
Macrophages	188	287	420	3.95	6.44	8.64
Monocytes	10	24	24	0.21	0.54	0.49
Neutrophils	28	46	25	0.59	1.03	0.51
NK cells	3	2	11	0.06	0.04	0.23
NKT	16	19	97	0.34	0.43	2.00
T cells	15	18	46	0.31	0.40	0.95
Tgd	2	5	6	0.04	0.11	0.12
NA	18	22	54	0.38	0.49	1.11

## Data Availability

RNAseq data described in this study have been deposited in the gene expression omnibus (GEO) under accession number GSE200005, and single-cell sequencing data are available in the NCBI GEO database under GSE200139.

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
