# Peer review of "Single-Cell Analyses of a Novel Mouse Urothelial Carcinoma Model Reveal a Role of Tumor-Associated Macrophages in Response to Anti-PD-1 Therapy"

_cancers, 2022, doi:10.3390/cancers14102511_

Round 1

Reviewer 1 Report

Comments for the authors:

Xu et al. have established a new organoid-based, syngeneic, murine model of urothelial carcinoma. They have also characterized the tumors further using IHC, IF, total RNA sequencing, and single-cell RNA sequencing. The TKO tumors show significant expression of urothelial markers and recapitulate the therapy response to anti-PD1 therapy akin to human patients. The manuscript is engaging and well-written. However, I have a few concerns about single-cell data and some of the claims made in the manuscript.

Major concerns:

  1. How stable is the TKO model? How long do the mice with TKO tumors survive? How long will the therapy window be available for the researchers who want to use this model? How cumbersome is the generation of the TKO model compared to other well-established bladder cancer models such as MB49 and BBN? Please provide this information in the discussion section.
  1. Line 374-375 and the title suggest that the manuscript revealed the role of tumor-associated macrophages in the anti-PD1 therapy. While it is interesting that there are more macrophages in the TKO tumors responding to the therapy, none of the experiments rule out the role of other immune cell populations such as T-cells, B-cells and NKT, etc. Please clarify why more stress was placed on the tumor-associated macrophages alone.
  1. Fig 4C and 4D, the data is captured using high-throughput technology and captures a limited number of the immune cells (~400- to 700 cells) and should be validated. The authors may perform multicolor flow cytometry to quantify the immune cells' frequency and absolute numbers in the TKO tumors. The flow data may provide a reliable comparative analysis of the immune cells among the three groups.
  1. The authors have used ScRNA seq data and emphasize the role played by mere frequency of the immune cell populations in the tumor responsiveness to anti-PD1 therapy. The authors have not provided any data on the transcriptional signatures of these cell types. Hence, it would be informative to perform differential expression and pathway enrichment analysis to understand the differences in the immune cell populations present in the responders vs. non-responders.
  1. The authors claim that the MB48 tumors are extremely heterogenous; please cite adequate references to support the claim. On the contrary, in Figures 3A and 3B, PCA projections and heatmap show similar transcriptional signatures from three MB49 tumors. Please explain the discrepancy.

Minor concerns:

  1. Fig 1c does not have a DNA ladder and the arrows indicating the sizes.
  2. Fig 1B scale bars are missing in 3 out of 4 microscopy images.

Reviewer 2 Report

General comments:

The treatment of muscle invasive bladder cancer still presents an unmet challenge. Therefore development of relevant immunocompetent mouse models that can reproduce features of invasive bladder cancer disease is highly relevant, especially in light of immune checkpoint therapies and other immunotherapies.

The paper present an approach to develop a preclinical model of a bladder cancer. They developed urothelial carcinomas from triple knockout organoids transplanted into immunocompetent

mice, with TKO tumors exhibiting basal subtype. The study demonstrates the ability of this model to study response to anti-PD-1 ICI therapy, with heterogeneous response.

The presented methodology is very relevant for field of bladder cancer research and showing promise to study the response to immunotherapies in a more relevant pre-clinical mouse model of bladder cancer. As one of the results using the developed model the study shows that responders had a higher percentage of infiltrated macrophages, but this should be put in contex of other studies that examine the mechanisms of resistance to ICI in non-responders.

Altogether the authors present  an interesting and promising approach.

Specific comments:

Methods

143: »Immunofluorescent (IF) staining and immunohistochemistry (IHC) were performed as previously described.« - please provide a reference.

Results

274: »The expression pattern of urothelial markers in intravesical tumors is identical to subcutaneous and orthotopic tumors (data not shown)« Please provide the images in the supplementary section.

Figure 2

Staining with urothelial biomarkers - was histology and immunostaining compared to human bladder tumors? It would be interesting to compare.

212 and Figure 4C.

In section 2.7. it is stated:

»SingleR [36] package was utilized to identify the immune

cell types using ImmGen reference dataset« and

»Further annotation using SingleR with ImmGen revealed cell populations of basophils, B cells, dendritic cells, macrophages, monocytes, neutrophils, NK cells, and T cells in anti‐PD‐1 treated xenografts (Figure 4C and Table 1).«

Please provide more details of how the cell populations of T cells, NK, macrophages were identified. Did authors validated single cell immunophenotyping with standard flow cytometry immunophenotyping – please provide additional data or references.

449 »Our results also suggest that modulating tumor‐associated macrophages may be important for optimizing ICI immunotherapy responses.«

Are there any observations in the literature supporting this observation?

For example tumor-associated macrophages were demonstrated to limit PD-1 blockade by removing anti-PD-1 antibodies from PD-1+ CD8+ T cells (Arlauckas et al. 2017, doi: 10.1126/scitranslmed.aal3604)

The authors should comment how their results is related to existing observations of the mechanisms of resistance to immune checkpoint inhibitors.

Reviewer 3 Report

This manuscript by Xu et al developed triple knockout (TKO: Trp53, Pten, Rb1) organoid-based urothelial carcinomas model and exhibited the derived bladder cancers after implantation recapitulated the molecular phenotypes and heterogeneous immunotherapy responses. The authors also compared the characteristics of the TKO organoid in vivo and in vitro to MB49 murine bladder cancer model. They confirmed TKO tumors as a basal subtype by RNAseq. Lastly they used scRNAseq to further explained potential mechanism for the responding status to anti-PD-1. They found out responder xenografts displayed significantly increased immune cell infiltration, especially more T cells and macrophages in comparison to non-responder xenografts. This is a well-written manuscript and there are several concerns need more clarification.

1). Please clarify if TKO mice with organ-specific Cre introduction generate urothelial carcinoma. If yes, please explain the reason that the authors did not use these models to compare organoid-based models in terms of characteristics and tumor behaviors.

2). scRNA data analysis in this paper is relatively superficial and the results derived from this data analysis can be replaced with flowcytometry. If scRNA data is preferred, it is recommended to have more biological repeat in order to have significant comparison and more in-depth analysis would be appropriate.

Reviewer 4 Report

The authors made triple knockout (TKO) urothelial organoids by treating cells collected from bladder tissue of triple floxed mice (Trp53f/f, Ptenf/f, Rb1f/f) with a mixed C57BL/6:129/Sv:FVB genetic background with Cre recombinase ex vivo. This organoid has evolved into a high-grade urothelial cancer cell (TKO tumor). When TKO tumor cells were inoculated into the bladder of C57 BL/6J mice, bladder tumors were formed. When anti-PD1 antibody was administered to mice bearing this bladder tumor, the number of T cells and macrophages infiltrating the bladder tumor increased when the tumor shrank.   I have a major concern. The TKO tumor cell should have a C57BL/6: 129/Sv: FVB genetic background. The Major Histocompatibility Complex (MHC) haplotype of C57BL/6 and 129/Sv is both b, but the MHC haplotype of FVB is q. Therefore, for C57BL/6 mice inoculated with TKO tumor cells in the bladder, TKO tumor cells also possessing MHC q are considered to be allogeneic rather than syngeneic.  

It seems that alllogeneic TKO tumor cells were rejected in immunocompetent C57BL/6 mice. I suspect this didn't work as a model.

Round 2

Reviewer 1 Report

Xu et have adequately addressed all my concerns. I recommend publication of the manuscript.

Reviewer 3 Report

The authors answered the critics appropriately.

Reviewer 4 Report

This article is acceptable.